# Multi-Modal Enhancement Transformer Network for Skeleton-Based Human Interaction Recognition

**DOI:** 10.3390/biomimetics9030123

**Published:** 2024-02-20

**Authors:** Qianshuo Hu, Haijun Liu

**Affiliations:** School of Microelectronics and Communication Engineering, Chongqing University, Chongqing 400044, China; huqianshuo@gmail.com

**Keywords:** transformer, skeleton data, human interaction recognition, hypergraph representation

## Abstract

Skeleton-based human interaction recognition is a challenging task in the field of vision and image processing. Graph Convolutional Networks (GCNs) achieved remarkable performance by modeling the human skeleton as a topology. However, existing GCN-based methods have two problems: (1) Existing frameworks cannot effectively take advantage of the complementary features of different skeletal modalities. There is no information transfer channel between various specific modalities. (2) Limited by the structure of the skeleton topology, it is hard to capture and learn the information about two-person interactions. To solve these problems, inspired by the human visual neural network, we propose a multi-modal enhancement transformer (ME-Former) network for skeleton-based human interaction recognition. ME-Former includes a multi-modal enhancement module (ME) and a context progressive fusion block (CPF). More specifically, each ME module consists of a multi-head cross-modal attention block (MH-CA) and a two-person hypergraph self-attention block (TH-SA), which are responsible for enhancing the skeleton features of a specific modality from other skeletal modalities and modeling spatial dependencies between joints using the specific modality, respectively. In addition, we propose a two-person skeleton topology and a two-person hypergraph representation. The TH-SA block can embed their structural information into the self-attention to better learn two-person interaction. The CPF block is capable of progressively transforming the features of different skeletal modalities from low-level features to higher-order global contexts, making the enhancement process more efficient. Extensive experiments on benchmark NTU-RGB+D 60 and NTU-RGB+D 120 datasets consistently verify the effectiveness of our proposed ME-Former by outperforming state-of-the-art methods.

## 1. Introduction

Human action recognition technology has been widely used in the fields of bionic robots, medical care, video surveillance, intelligent transportation, and human–robot interaction [1,2,3,4,5,6,7,8]. Skeletons are made up of a variety of different shapes, with complex internal and external structures, which play a role in supporting and protecting the human body. As a biological feature of the human body, the skeleton data can fully represent the kinetic features of human actions. Compared to RGB and depth data, skeleton data are more robust than RGB and depth data to illumination change and viewpoint occlusion. Advances in depth cameras and pose estimation methods [9,10,11,12] have also made it easier for bionic robots to acquire skeleton data.

Previous studies [13,14,15] have confirmed that second-order information about the skeleton plays a complementary role in action recognition. The second-order information about the skeleton, which could also be called the skeletal modality, included joint coordinates, bone vector, joint coordinate motion, and bone vector motion. An early approach to multi-modal fusion strategy, known as back-end fusion, was extremely time-consuming and could not be trained in an end-to-end way. Diverse modalities of skeleton data were input into the model separately to obtain partial results. Then, the partial results were manually fused to obtain the final recognition result. Song et al. [16] proposed a training framework called an input branch network, which allowed the skeleton features of each modality to be trained separately in the branch network, and then concatenated the features from different modalities in a specific layer to continue training using the main stream network. Although this method improved the training efficiency to a certain extent, it had one disadvantage: in the input branch network, there is no information transfer between different skeletal modalities. The absence of information transfer meant that every specific skeletal modality could not learn complementary features from other skeletal modalities. Furthermore, the data distribution of skeleton features after concatenation was considerably different, which hindered the subsequent recognition.

In order to solve the problem of missing information transfer, we utilize the cross-attention mechanism to fuse skeleton features from different modalities. Through successive training iterations, the cross-attention mechanism can progressively enhance the skeleton features of a specific modality by learning the directional pairwise attention between the specific modality and other skeletal modalities. However, the conventional cross-attention mechanism can only handle the features of two modalities. Inspired by the human visual neural network, which consists of multiple subnets [17] where different neural subnets are connected by dendrites in the brain cortex [18], we propose to add a global context subnet to interact with other skeletal modalities. Specifically, our improvement is to build a context progressive fusion block to save and update the global context, and then use the global context to interact with each skeletal modality. The global context is gradually transformed from low-level to high-level with network training. During this process, the global context aggregated sufficient information from different skeletal modalities.

Although Transformer had shown advantages in multi-modal enhancement, it worked poorly with structured data, such as skeleton data. The main reason was that each joint in the skeleton topology contained particular spatial structural information, which is not perceived by the vanilla Transformer. Ying et al. [19] and Zhou et al. [20] proposed two methods for embedding the spatial structural information about the skeleton into the Transformer. They were relative position embedding (RPE) and hypergraph embedding, which embed the pairwise and higher-order relations of skeletal joints into the Transformer, respectively. However, the above methods were used for single-person action recognition and were not suitable for two-person interaction recognition.

To solve the limitations of the existing skeletal topology, we design a simple yet effective two-person skeleton topology and hypergraph. We first consider the skeleton topology of two persons as a whole and then connect the limbs and heads of two persons to obtain the two-person skeleton topology. Then, we use a learnable method to automatically convert the two-person skeleton topology into a two-person hypergraph representation. By embedding these two kinds of spatial structure information into the Transformer, the model can better recognize the interaction information about the two people.

In this paper, we propose a multi-modal enhancement transformer (ME-Former) network, which can enhance different skeletal modalities simultaneously and recognize two-person interaction. ME-Former includes a multi-modal enhancement module (ME) and a context progressive fusion block (CPF). More specifically, each ME module consists of a multi-head cross-modal attention block (MH-CA) and a two-person hypergraph self-attention block (TH-SA). The MH-CA block is responsible for utilizing the global context generated from the CPF block to enhance the skeleton features of specific modalities. The TH-SA block can encode the two-person skeleton topology and two-person hypergraph representation as structural information and embed them into self-attention to better learn two-person interaction. On the other hand, the CPF block performing the information interaction among all the skeletal modalities is proposed to generate and update the global context.

The strategy of our research is as follows: Section 1 introduces the research background of this work and our solutions to existing two problems. Section 2 describes the related work, including skeleton-based action recognition methods and human interaction recognition methods. Section 3 recaps the prior knowledge involved in this work. Section 4 states the network structure and implementation details of the ME-Former. Section 5 evaluates the validity of our proposed ME-Former and compares it with other advanced methods. Section 6 summarizes this paper and predicts the future development direction. The main contributions of our work are as follows:A novel multi-modal enhancement transformer (ME-Former) network is proposed for skeleton-based human interaction recognition, overcoming the challenge of different skeletal modalities not being able to effectively utilize complementary features;We propose a simple yet effective two-person skeleton topology and a two-person hypergraph representation, which can model the pairwise and higher-order relations between human joints, and use the TH-SA block to embed the two kinds of structural information into the ME-Former for better recognizing human interaction;The extensive experimental results highlight the benefits of our multi-modal enhancement transformer and two-person hypergraph representation. Our proposed ME-Former outperforms state-of-the-art methods on two different skeleton-based action recognition benchmarks.

## 2. Related Work

### 2.1. Skeleton-Based Action Recognition

With respect to methods for modeling human skeleton data, skeleton-based action recognition methods can be simply categorized into convolutional neural networks (CNNs), recurrent neural networks (RNNs), GCNs, and Transformers. CNNs process the skeleton data as a pseudo-image [21,22,23,24], while RNNs flattens the skeleton data into a sequence [25,26,27,28]. Nevertheless, the above processing methods lose the semantic information about the skeleton data and the higher-order relations of the joints.

**GCN-based approaches.** GCNs modeled the skeleton data as a graph [29,30], in which the nodes are joints and the edges are bones. It turned out that GCNs were better at modeling the spatial features of skeleton data. Yan et al. [31] proposed ST-GCN which first extracts and aggregates the spatio-temporal information about the skeleton using the topological diagram, whereas ST-GCN used a handcrafted skeleton graph, which cannot characterize the relevance of joints that are not naturally connected, limiting the ability of GCNs to extract and aggregate features. To improve the capacity of GCNs, Shi et al. [13] proposed 2s-AGCN which included the second-order information (the lengths and directions of bones) of the skeleton data in the range of feature extraction, and used two streams to process the information about different modalities and integrate it. The model constructed an adaptive topology map, which can be updated automatically with the backpropagation algorithm of the neural network to characterize the node connection strength better. None of the above models considered that each channel can express richer potential action features until CTR-GCN [32] modeled channel-wise topologies by dividing the channels into groups and only sharing the skeleton topology between groups. CTR-GCN learned a shared topology and channel-specific correlations simultaneously to enhance the feature extraction and aggregation ability of GCNs without significantly increasing the parameters. Lee et al. [33] proposed HD-GCN which decomposes the human skeleton graph into six hierarchical subsets, highlighting the distance relationship between joints. More importantly, it allowed nodes, that are not naturally connected, to be connected in a sub-graph, which is better for transmitting information between the joints. One disadvantage of GCNs was their inability to model long-distance spatio-temporal dependence due to the local nature of graph convolution.

**Transformer-based approaches.** Transformer was a novel deep-learning model, which gradually appeared in the field of action recognition because of its strong capacity for parallel processing sequence data and modeling long-term dependency. Transformers focused primarily on modeling spatio-temporal correlations and embedding semantic information specific to the skeleton joints. Wang et al. [34] proposed a Transformer-based network (IIP-Transformer) that models the intra-part and inter-part dependencies on the spatial dimension, enabling a more detailed representation of features. Plizzari et al. [35] proposed a novel Spatial–Temporal Transformer network (ST-TR) in which the spatial self-attention module and temporal self-attention module are used to capture the correlation between different nodes in a frame and the dynamic relationship between the same node in the whole frames. Zhang et al. [36] proposed a Spatial-Temporal Specialized Transformer (STST), which is composed of the spatial transformer block and directional temporal transformer block. The spatial transformer block was designed to model the dependency between joints and semantic representation in each frame separately, and the directional temporal transformer block was designed to model the long-term temporal dependency with a direction-aware strategy. Ying et al. [19] proposed a Graphformer to incorporate the joint connection information into attention computation via a distance-based relative positional embedding method. The natural human topology graph contained only two joints on each edge and this pairwise structure limited the representation of higher-order kinematic dependencies. Zhou et al. [20] further proposed a Hypergraph Transformer. The hypergraph contained hyperedges that are capable of connecting two or more vertices. By embedding hypergraph representations into attention calculations, the Transformer could learn high-order relations of skeleton joints.

### 2.2. Human Interaction Recognition

Human interaction recognition is a sub-task of action recognition. The methods of interaction recognition need to learn both individual information and interactive information between individuals. Ji et al. [37] proposed an interactive body part model. The model split the human body into five body parts and combined the limbs of two participants into eight interacting body part pairs. For each pair, they calculated the joint feature in three frames, called poselets. Yang et al. [38] introduced a pairwise graph that connects the joints of two people in pairs in a predefined way. Both Li et al. [39] and Zhu et al. [40] explicitly modeled the interactive relationship of the two persons into an additional inter-body matrix, and took the intra-body matrix and inter-body matrix in parallel into the graph convolution network to obtain the results. Perez et al. [41] proposed a two-stream interaction relation network called LSTM-IRN which models the internal joint relationship of the same subject and the inter-relationship of different subjects. Furthermore, they defined structured pairwise operations to extract and learn valuable extra information (distance and motion) from each pair of joints. Pang et al. [42] proposed a novel Interaction Graph Transformer (IGFormer) network to achieve skeleton-based interaction recognition by modeling interacting body parts as graphs. More specifically, IGFormer constructed interaction graphs based on semantic and distance correlations between the body parts of interactive persons and aggregated information about interacting body parts by the learned graph. Gao et al. [43] first treated the topology of two persons as a whole and used attention-driven modules to capture interactive correlations dynamically. Li et al. [44] designed several graph labeling strategies to accelerate the convergence of the model. In addition, they integrated the data pre-processing methods and developed a symmetry processing method to reduce the high intra-class variation caused by the order of people.

## 3. Preliminaries

In this section, we recap the definition of attention mechanism and hypergraph representation.

### 3.1. Attention Mechanism

The full name of attention is scaled product attention. The function of scaled dot-product attention is to map a query and a set of key-value pairs to an output, where the query (*Q*), key (*K*), value (*V*), and output are all vectors. The attention values are generated by applying the softmax function to the dot product of *Q* and *K* and dividing each element by $\sqrt{d_{k}}$. The attention value is multiplied by *V* to get a weighted output:(1)$$\mathrm{Attention}\left(Q,K,V\right)=\mathrm{SoftMax}\left(\frac{QK^{T}}{\sqrt{d_{k}}}\right)V.$$

It is worth noting that we added a multi-head attention mechanism in practice to improve the performance of the Transformer. This divides the channel dimension into groups and applies attention to each group in parallel to efficiently learn distinctive features. The multi-head attention is calculated as follows:(2)$$\mathrm{MultiHead}\left(Q,K,V\right)=\mathrm{Concat}\left(head_{1},head_{2},\ldots ,head_{h}\right)W^{O},$$
(3)$$head_{i}=\mathrm{Attention}\left(QW_{i}^{Q},KW_{i}^{K},VW_{i}^{V}\right),i\in \left[1,h\right],$$
where parameter matrices $W^{O}\in R^{C\times d_{k}}$, $W_{i}^{Q}\in R^{h\times C_{head}\times d_{q}}$, $W_{i}^{K}\in R^{h\times C_{head}\times d_{k}}$, and $W_{i}^{V}\in R^{h\times C_{head}\times d_{v}}$. *h* is the number of heads and $C_{head}=C/h$ is the number of channels per head. In this work, we employ h=8.

### 3.2. Hypergraph Representation

In a standard graph, only two vertices can be connected to each edge. The hypergraph is not restricted and the hyperedge can connect more than two vertices. A hypergraph is defined as G=(V,E,W) which consists of a vertex set V, a hyperedge set E, and a symmetric matrix W which contains hyperedge weights. The hypergraph G can be denoted by a |V|×|E| incidence matrix *H*, with entries defined as follows:(4)$$h\left(v,e\right)=\left\{\begin{array}{c} 1\;if\;v\in e\\ 0\;if\;v\notin e. \end{array}\right.$$

The degree of a vertex v∈V is defined as $d\left(v\right)=\sum_{e\in E}w\left(e\right)h\left(v,e\right)$ and the degree of a hyperedge e∈E is defined as $d\left(e\right)=\sum_{v\in V}h\left(v,e\right)$. We use diagonal matrices $D_{v}$ and $D_{e}$, respectively, to represent the degree matrix of vertices and hyperedges, where the elements on the diagonal are the degrees of each vertex or hyperedge.

## 4. The Proposed Method

In this section, we introduce the whole structure of our multi-modal enhancement transformer network (ME-Former) which is depicted in Figure 1. The ME-Former network consists of two stages: the feature extraction and fusion stage, and the feature refinement stage.

At the feature extraction and fusion stage, the ME-former comprises five parallel data streams. They are four skeletal modalities and the global context. The four skeletal modalities are joint coordinates (J), bone vector (B), joint motion (JM), and bone motion (BM). Each specific modal stream passes through the multi-modal enhancement module (ME) and multi-scale temporal convolution block (MS-TC) in one layer, for a total of four layers. The ME module has two functions: (1) Enhance and complement specific skeletal modalities with the global context. (2) Model long-distance spatial dependencies on each skeletal modality itself. We adopt the MS-TC block from [45] to model the temporal dependence of every specific modality. The global context, which is generated by the context progressive fusion block (CPF), aggregates information about the four skeletal modalities in each layer. The global context is used to enhance each specific modality so that the skeleton features from different modalities are gradually fused together. After four layers of feature extraction and fusion, the four kinds of skeleton features enhanced by the feature extraction and fusion stage are concatenated together to be fed into the feature refinement stage.

At the feature refinement stage, the ME-Former treats the fused features as a unified skeletal modality. The feature refinement stage has six layers; each layer uses TH-SA blocks and MS-TC blocks to model spatial and temporal skeleton features. Finally, the action prediction scores are obtained by feeding skeleton features into the global average pooling layer and the fully connected layer.

### 4.1. Modality Enhancement Module

The ME module is placed in each layer of the feature extraction and fusion stage. It can not only extract the spatial features of the specific modality but also utilize the global context to gradually enhance the skeleton features of each specific modality. The technical details of the ME module are shown in Figure 2. The inputs to the ME module are the global context information $Y_{G}^{\left[i\right]}$ and the four skeleton modalities $Y_{*}^{\left[i\right]}$, where ∗∈{J,B,JM,BM}, which act as the source and target modality, respectively. The source modality is used to enhance the target modality; this enhancement process can be denoted as source→target. The final output $Y_{*}$ of the ME module can be expressed as follows:(5)$$Y_{*}^{\left[i+1\right]}={\mathrm{ME}}_{G\to *}^{\left[i\right]}\left(Y_{G}^{\left[i\right]},Y_{*}^{\left[i\right]}\right),$$
where ∗∈{J,B,JM,BM} and *i* represents the i-th layer of the ME-Former. The ME module does not change the feature shape of skeletal modalities; the size of both the input and output are the same, $Y_{*}\in R^{N\times C\times T\times J}$, where *N* represents batch size, *C* represents channel number of features, *T* represents sequence length of the skeleton frames, and *J* represents the number of human body joints.

Specifically, the ME module consists of a two-person hypergraph self-attention block (TH-SA), which is responsible for modeling the spatial dependence of specific modal skeleton features, and a multi-head cross-modal attention block (MH-CA), which is responsible for enhancing the skeleton features of a specific modality with global context information. The intermediate outputs of the two blocks are computed as follows:(6)$$Y_{*}^{\left[i\right]}=\text{TH-SA}(\mathrm{LN}\left(Y_{*}^{\left[i\right]}\right),$$
(7)$$Y_{G\to *}^{\left[i\right]}=\mathrm{MultiHead}\left(\mathrm{LN}\left(Y_{*}^{\left[i\right]}\right),\mathrm{LN}\left(Y_{*}^{\left[i\right]}\right),\mathrm{LN}\left(Y_{G}^{\left[i\right]}\right)\right),$$
where LN represents the layer normalization operation. For a clear description, we first introduce the enhancement process of the specific modality. The detailed implementation of the TH-SA block is left at the end of this section. Then, the intermediate output of the two blocks is processed via the following dynamic filter:(8)$$W_{*}^{\left[i\right]}=\mathrm{Sigmoid}\left(\text{Conv2d}\left(Y_{*}^{\left[i\right]}\right)+\text{Conv2d}\left(Y_{G\to *}^{\left[i\right]}\right)\right),$$
(9)$$Y_{*}^{\left[i\right]}=W_{*}^{\left[i\right]}⊙Y_{G\to *}^{\left[i\right]}+\left(1-W_{*}^{\left[i\right]}\right)⊙Y_{*}^{\left[i\right]}.$$

The weight of each branch can be dynamically determined by learnable parameters of the Conv2d function. This operation can filter out features that are not important in the uni-modal enhancement process. Finally, we use a simple convolution layer with residual connection to obtain the output of ME module $Y_{*}^{\left[i+1\right]}$.

### 4.2. Context Progressive Fusion Block

The context progressive fusion block (CPF) aggregates the skeleton features of each specific modality in the current layer and transforms them into a global context for cross-modal information interacting with the ME modules of the next layer. This is more efficient than making different skeletal modalities exchange information in pairs. The technical details of the CPF block are shown in Figure 3. The input to the CPF block is the global context $Y_{G}^{\left[i\right]}$ of the current layer and four skeleton modalities $Y_{*}^{\left[i\right]}$, where ∗∈{J,B,JM,BM}.

They are first converted to the features $z_{J}$, $z_{B}$, $z_{\mathrm{JM}}$, $z_{\mathrm{BM}}$, and $z_{G}$ with a range [−1, 1] by the Conv2d and Tanh functions. The magnitude of the feature values of different skeletal modalities also defines their importance. Next, we apply the softmax function to these features to get the weight for each skeletal modality. We then multiply the features of each modality with the corresponding weights and add them together to obtain global context $Y_{G}^{\left[i+1\right]}$. The calculation process is formulated as follows:(10)$$z_{*}=\mathrm{Tanh}\left(\text{Conv2d}\left(Y_{*}^{\left[i\right]}\right)\right),$$
(11)$$w_{*}=\frac{\exp(z_{*})}{\exp\left(z_{J}\right)+\exp\left(z_{B}\right)+\exp\left(z_{\mathrm{JM}}\right)+\exp\left(z_{\mathrm{BM}}\right)+\exp\left(z_{G}\right)},$$
(12)$$Y_{G}^{\left[i+1\right]}=Y_{J}^{\left[i\right]}⊙w_{J}+Y_{B}^{\left[i\right]}⊙w_{B}+Y_{JM}^{\left[i\right]}⊙w_{JM}+Y_{BM}^{\left[i\right]}⊙w_{BM}+Y_{G}^{\left[i\right]}⊙w_{G},$$
where ∗∈{J,B,JM,BM}, ⊙ means Hadamard Product, and exp means Exponential Function. This process is a variant of attention, which fuses important features from different skeletal modalities and the global context of the current layer into the global context of the next layer. Finally, we feed global context $Y_{G}^{\left[i+1\right]}$ into a convolution layer with residual connections to obtain the output of the CPF block.

### 4.3. Two-Person Hypergraph Self-Attention Block

TH-SA blocks exist in both the two stages of the ME-Former network. Its function is to model the spatial dependencies of skeleton features. To better model spatial features, TH-SA embeds two kinds of skeleton structure information based on vanilla SA, namely two-person skeleton topology and two-person hypergraph representation. They indicate pairwise relations and higher-order relations between skeleton joints, respectively. The pipeline of the TH-SA block is shown in Figure 4.

Conventional separate single-person skeleton topologies cannot capture interaction information between two persons, so we design a simple yet effective two-person skeleton topology. The difference between two separate single-person skeleton topologies and our designed two-person skeleton topologies is that we consider the skeleton topology of two persons as a whole and then connect the limbs and heads of the two persons. Skeleton features that belong to two persons can interact through these additional connections by the skeleton topology embedding method. Their visualization is shown in Figure 5a,b.

The attention value *A* consists of three components, which are vanilla SA, skeleton topology embedding, and two-person hypergraph (2p-Hypergraph) embedding. Given the input $Q=K=V=Y_{*}^{\left[i\right]},$ where ∗∈{J,B,JM,BM}, the attention value is calculated as follows:(13)$$A=QK^{T}+QR_{k}^{T}+QE_{hyper}^{T}.$$

Skeleton topology embedding and 2p-hypergraph embedding are realized by multiplying structural information $R_{k}$ and 2p-hypergraph features $E_{hyper}$ with skeleton features, respectively. The detailed procedure for obtaining $R_{k}$ and $E_{hyper}$ is in Section 4.3.1 and Section 4.3.2. Both embedding methods are designed to make the transformer aware of the specific spatial semantics of each human joint. By matrix multiplication of the attention value *A* with *V*, the output of TH-SA can be obtained as $Y_{*}^{\left[i\right]}$.
(14)$$Y_{*}^{\left[i\right]}=softmax\left(\frac{A}{\sqrt{d_{k}}}\right)V.$$

#### 4.3.1. Encoding Structural Information about Two-Person Skeleton Topology

The joints of the human body are naturally connected by bones, forming a bio-mechanical model. The movement of each joint in an action has a chain effect on the other joints. Therefore, it is necessary to embed the structural information about the human body into each position of skeleton features.

We refer to the existing relative positional embedding (RPE) method and propose a K-hop RPE method suitable for skeleton data. We first obtain the set $P\in R^{J\times J}$ using the Shortest Path Distance (SPD) algorithm [19], where $p_{ij}$ indicates the distance or number of hops between joint *i* and joint *j*. Then, we define a learnable parameter $W_{k}\in R^{k\times C}$ to represent the weights corresponding to different distances (max hop is k). Finally, each element in set *P* is assigned the corresponding weight in $W_{k}$ to obtain the structural information about skeleton topology $R_{k}\in R^{J\times J\times C}$.
(15)$$R_{k}=\sum_{i=0}^{J}\sum_{j=0}^{J}W_{k}\left[p_{i,j}\right].$$

#### 4.3.2. Deriving Two-Person Hypergraph Feature

The definition of hypergraph has been explained in Section 3.2. We set the degree of each vertex d(v)=1 for efficient realization. In this particular case, the human joints can be divided into |E| discrete subsets and the incidence matrix *H* is equivalent to a partition matrix. Each row is a one-hot vector denoting the group to which each joint belongs. So how to formulate a partition strategy to obtain hypergraph representation is crucial.

We demonstrate two common skeleton topology partition strategies based on body prior knowledge in Figure 5c,d. The body parts partition strategy divides the skeletal topology into four limbs and a trunk, while the upper and lower partition strategy divides the skeletal topology into upper and lower halves. Although the above partitioning strategies can reflect the relations between multiple joints to a certain extent, these manual methods are not the optimal solution and have poor generalization. Unlike these, in the TH-SA block, we use a learnable partition matrix that can automatically transform a two-person skeleton topology into a two-person hypergraph representation.

Although the binary incidence matrix $H\in R^{\left|V|\times |E\right|}$ can be regarded as a partition matrix, its values are discrete. We parameterize it to make values continuous by applying a softmax function along the column axis:(16)$$\tilde{H}=\left\{{\tilde{h}}_{ve}=\frac{\exp(h_{ve})}{\sum_{e=1}^{\left|E\right|}\exp\left(h_{ve}\right)}:v=1\ldots |V|,e=1...|E|\right\},$$Continuous partition matrix $\tilde{H}$ can be optimized jointly with other Transformer parameters. At the end of the optimization, we re-convert the values of the partition matrix from continuous to discrete by applying the argmax function along each row of $\tilde{H}$:(17)$$\bar{H}=\mathrm{argmax}\left(\tilde{H}\right),$$The visualization of the learned partition matrix $\bar{H}$ is shown in Figure 5e.

Then, we utilize the $\bar{H}$ to get a two-person hypergraph representation and embed it into the skeleton features. Given the partition matrix $\bar{H}\in R^{\left|V|\times |E\right|}$ and skeleton joint features $X\in R^{|V|\times C}$, the representation of hyperedge set $E\in R^{|E|\times C}$ can be calculated by the following formula:(18)$$E=D_{e}^{-1}{\bar{H}}^{T}XW_{e},$$
where the inverse degree matrix $D_{e}$ of hyperedges plays the role of normalization and $W_{e}$ is a learnable parameter matrix. Finally, we assign the weight of each hyperedge to the corresponding subsidiary joints to get the two-person hypergraph feature $E_{hyper}\in R^{|V|\times C}$:(19)$$E_{hyper}=HD_{e}^{-1}{\bar{H}}^{T}XW_{e}.$$

## 5. Experiment

### 5.1. Datasets

**NTU-RGB+D 60 (NTU-60)** [46] is a large-scale dataset for RGB+D human action recognition with more than 56 thousand video samples and 4 million frames, collected from 40 distinct subjects. It contains 60 different action classes including daily, mutual, and health-related actions. There are two standard evaluation protocols for this dataset including the cross-subject (X-Sub) protocol in which training data comes from 20 subjects and testing data comes from the other 20 subjects, and the cross-view (X-View) protocol where two cameras are used for training and the third one is used for testing. NTU-60 dataset contains 11 two-person interaction classes involving ‘punch/slap’, ‘pat on the back’, ‘giving object’, ‘walking towards’, ‘kicking’, ‘point finger’, ‘touch pocket’, ‘walking apart’, ‘pushing’, ‘hugging’, and ‘handshaking’; we denote this part of the dataset as **NTU-60*** in the following experiments.

**NTU-RGB+D120 (NTU-120)** [47] is a large 3D joint dataset of human movements. This dataset expands the NTU-RGB+D dataset by adding 57,367 skeletal sequences and 60 additional action categories. This dataset was captured with three cameras, with a total of 113,945 samples, by 106 volunteers. The dataset was also partitioned between the training and testing sets using two different partitions. (1) Cross-subject (X-Sub): the training data is from 53 subjects and the test set is from the remaining subjects. (2) Cross-setup (X-Setup): half of the setups are used for training and the remaining ones are used for testing. This dataset contains 15 additional human interaction classes including ‘hit with object’, ‘wield knife’, ‘knock over’, ‘grab stuff’, ‘shoot with gun’, ‘step on foot’, ‘high-five’, ‘cheers and drink’, ‘carry object’, ‘take a photo’, ‘follow’, ‘whisper’, ‘exchange things’, ‘support somebody’, and ‘rock–paper–scissors’, resulting a total of 26 interaction classes; we denote this part of the dataset as **NTU-120*** in the following experiments.

### 5.2. Implementation Details

We set the batch size to 16 and resize the frame number of a single sample to 64; the other data pre-processing follows [44,48]. Our model has ten layers and the number of channels in each layer is 192. The attention mechanism uses eight heads. ME-Former is trained by Stochastic Gradient Descent (SGD) with momentum set to 0.9, and weight decay set to 0.0005. The total number of training epochs is 140 with the first 5 epochs to warm up [49]. Our base learning rate (LR) is 0.1, the LR decay rate is 0.1, and the decay step is [110, 120]. We set standard cross-entropy as a loss function. All experiments are conducted on the Pytorch deep-learning framework with two NVIDIA GeForce RTX 3090 GPUs and 64G RAM.

### 5.3. Comparison with State-of-the-Art Methods

We compare the recognition accuracy of our proposed ME-Former with other state-of-the-art methods on the NTU-60* and NTU-120* datasets. We categorize the methods into three types and rank them by publication time and accuracy. The results are shown in Table 1.

The first category includes RNN-based methods for human interaction recognition. Although RNN-based methods can model the interactive information about two persons to a certain extent, it is difficult to capture the skeleton’s pairwise connection information and semantic information with them. Under the X-Sub and X-View protocols of the NTU-60*, the recognition accuracy of our proposed method ME-Former is 4.87% and 4.10% higher than the latest LSTM-IRN model, respectively.

The second category includes GCN-based methods for human interaction recognition. GCN-based methods mainly capture interactive information by modeling additional topologies and are generally better than the former methods. Their common disadvantage is poor generalization and requires a lot of extra computational overhead. At the same time, the skeleton topology cannot properly represent the connection information about the various skeletal modalities that are gradually merging. Under the X-Sub protocols of the NTU-60*, the recognition accuracy of our model is 0.88% higher than the EfficientGCN-B1 and 0.03% higher than the 2s-AIGCN, respectively. Under the X-Sub protocols of the NTU-120*, the recognition accuracy of our model is 0.20% higher than the EfficientGCN-B1 and 0.13% higher than the 2s-AIGCN, respectively.

The third category includes Transformer-based methods for human interaction recognition. Transformer can model long-distance dependency, which has advantages over GCNs that can only extract local features. Transformer-based methods are lightweight and easier for multi-modal fusion. After embedding the spatial structural information about the skeleton, Transformer-based methods gradually became mainstream. Under the X-Sub and X-View protocols of the NTU-120*, the recognition accuracy of our model is 5.44% and 4.83% higher than the IGFormer, and 0.56% and 0.20% higher than the STSA-Net, respectively.

To further demonstrate the performance of our approach, we present the normalized confusion matrix of the EfficientGCN-B1 model (baseline) and our proposed ME-Former model on the NTU-60* (X-Sub) benchmark in Figure 6. The confusion matrix contains eleven action classes, and we can observe that our model outperforms the EfficientGCN model on almost all action classes except the actions ‘hugging’ and ‘pat on the back’. The most significant improvements in accuracy are ‘pocket touches’ and ‘kicking’, which improved 2.18% and 2.9% compared to baseline, respectively. The most difficult actions to recognize in the NTU-60* (X-Sub) dataset are ‘punch/slap’ and ‘giving object’, and we believe that better recognition of two-person interactions requires more detailed modeling of hand gestures. Benefiting our proposed methods, the ME-Former model has outperformed state-of-the-art RNN-based, GCN-based, and Transformer-based methods in human interaction recognition.

### 5.4. Ablation Studies

In this section, we conduct extensive ablation studies on the NTU-60* dataset using the cross-subject (X-Sub) protocol to validate the effectiveness of the proposed multi-modal enforcement transformer network (ME-Former). To prove the validity of our proposed two-person skeleton topology and two-person hypergraph partition strategy, we further conduct detailed ablation experiments on the components in the TH-SA block. The visualizations of two skeleton topologies and three partition strategies are shown in Figure 5.

We set the EfficientGCN [16] as our baseline, which consists of two stages: the feature extraction and fusion stage, and the feature refinement stage. EfficientGCN uses spatial graph convolution (SGC) blocks to model the spatial features of the skeleton data in both stages. Since there are four skeletal modalities (four input branches), four independent SGC blocks are required in the feature extraction and fusion stage. It is worth noting that both our ME-Former and the baseline use the same multi-scale temporal convolution block (MS-TC) for modeling temporal features of the skeleton data, so we omit it for clarity in the ablation studies.

#### 5.4.1. Effectiveness of Multi-Modal Enforcement Transformer

ME-Former adopts a similar framework to the baseline model; the difference is that ME-Former uses the TH-SA blocks to model the spatial features of the skeleton data in both parts. ME-Former adds two additional units to the input branch, enhancing each single modality by interchanging information with the other skeletal modalities. They are the multi-modal enforcement module (ME) and the context progressive fusion block (CPF). The ME module comprises a multi-head cross-modal attention block (MH-CA) and a two-person hypergraph self-attention block (TH-SA). We will gradually replace and add our methods to the baseline model to demonstrate the effectiveness of our proposed methods. The overall ablation results of the multi-modal enforcement transformer network (ME-Former) are shown in Table 2.

We first replace the SGC block in the baseline model with four TH-SA blocks to test the effect of the TH-SA block. We can observe that the recognition accuracy is improved by 0.49% but the number of parameters in the model is also greatly increased. This is because the baseline is a lightweight model with a lot of fine-tuning in the model structure.Next, we add MH-CA blocks to each input branch of the feature extraction and fusion stage, so that MH-CA and TH-SA are integrated into the ME modules. The recognition accuracy improves by 0.28%, which strongly demonstrates the significance of cross-modal information transfer between different input branches. Features of the other skeletal modalities can indeed enhance the skeleton features of each single modality.Finally, we add the CPF block to the feature extraction and fusion stage, and the result shows that the accuracy improves by 0.11%. This proves that using the global context can better aggregate the features among the various skeleton modalities and facilitate the enhancement of each single modality.

The results of all the ablation experiments prove that our proposed multi-modal enhancement transformer network is advanced and its recognition accuracy is 0.90% higher than the baseline model.

#### 5.4.2. Effectiveness of Two-Person Hypergraph Self-Attention Block

To eliminate distractions, we test the effects of each component of the TH-SA block when both the two stages of ME-Former only have the TH-SA block. As shown in Table 3, the results prove the effects of the two extra components in the TH-SA block. They are skeleton topology embedding and two-person hypergraph embedding.

We first compare the effect between single-person skeleton topology embedding and two-person skeleton topology embedding, showing that the performance of the latter is 0.21% higher than the former. This result proves that additional connections in the two-person skeleton topology facilitate learning interactive information between two persons. Adding two-person skeleton topology embedding improves accuracy by 1.18% compared to vanilla self-attention (SA).Then we add two-person hypergraph embedding to the vanilla SA and the results show that they improve the accuracy by 0.70%. The effect of two-person hypergraph embedding is slightly better than that of skeleton topology embedding, indicating that the higher-order relationship between joints can better reflect the difference between different actions than the pairwise relationship.

The full TH-SA is 1.61% higher than the vanilla SA while the number of parameters does not increase significantly. This proves that it is necessary to embed skeleton-specific connection information into the transformer.

In addition, we verify the influence of different partitioning strategies for skeleton topology in Table 4 and the results demonstrate the superiority of the two-person hypergraph. Our proposed two-person hypergraph partition strategy is 0.45% and 0.34% higher than the upper and lower body partition strategy, and body parts partition strategy, respectively. The two-person hypergraph partition strategy that can be learned automatically not only has higher recognition performance but also has stronger generalization.

## 6. Conclusions

In this work, we propose a multi-modal enhancement transformer network (ME-Former) for skeleton-based human interaction recognition. The ME-Former framework consists of two main components: the multi-modal enhancement module (ME) and the context progressive fusion block (CPF). The ME module is composed of a multi-head cross-modal attention block (MH-CA) and a two-person hypergraph self-attention block (TH-SA). These blocks have distinct roles in enhancing specific skeletal modalities and capturing long spatial dependencies between joints using a specific modality, respectively. On the other hand, the CPF block allows for information transfer between multiple skeletal modalities. The CPF block aggregates the skeleton features of different modalities and transforms them into a global context for multi-modal enhancement. With our proposed methods, ME-Former not only can efficiently fuse information from different skeletal modalities but also better recognize two-person interactions. Extensive experiment results show that our model has advanced performance on the NTU-60 and NTU-120 datasets. We believe that information transfer between different modalities can be more efficient and recommend further research into the feature fusion process. We expect that our proposed multi-modal enhancement transformer network will be applied in bionic robots and human-robot interaction interfaces. 

## Figures and Tables

**Figure 1 biomimetics-09-00123-f001:**
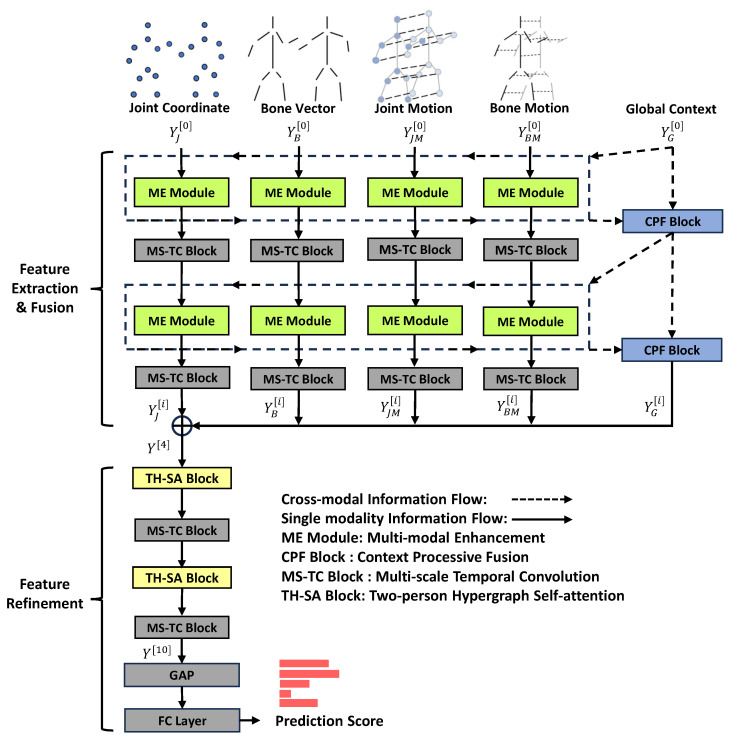
The overall architecture of our proposed multi-modal enhancement transformer network (ME-Former). The ME-Former network consists of two stages: the feature extraction and fusion stage, and the feature refinement stage. Each modality in the feature extraction and fusion stage has an independent information flow to learn distinctive spatio-temporal features. At the same time, each layer in the feature extraction and fusion stage also has a cross-modal information flow. The CPF block converts the features of all skeletal modalities in the previous layer into a global context. Then, it transfers cross-modal information into four ME modules in the current layer to enhance four single modalities. At the end of the feature extraction and fusion stage, ME-Former fuses the four skeletal modalities into a unified modality and models its spatial and temporal features. Finally, the skeleton features are sent into the fully connected layer (FC) to obtain the predicted scores.

**Figure 2 biomimetics-09-00123-f002:**
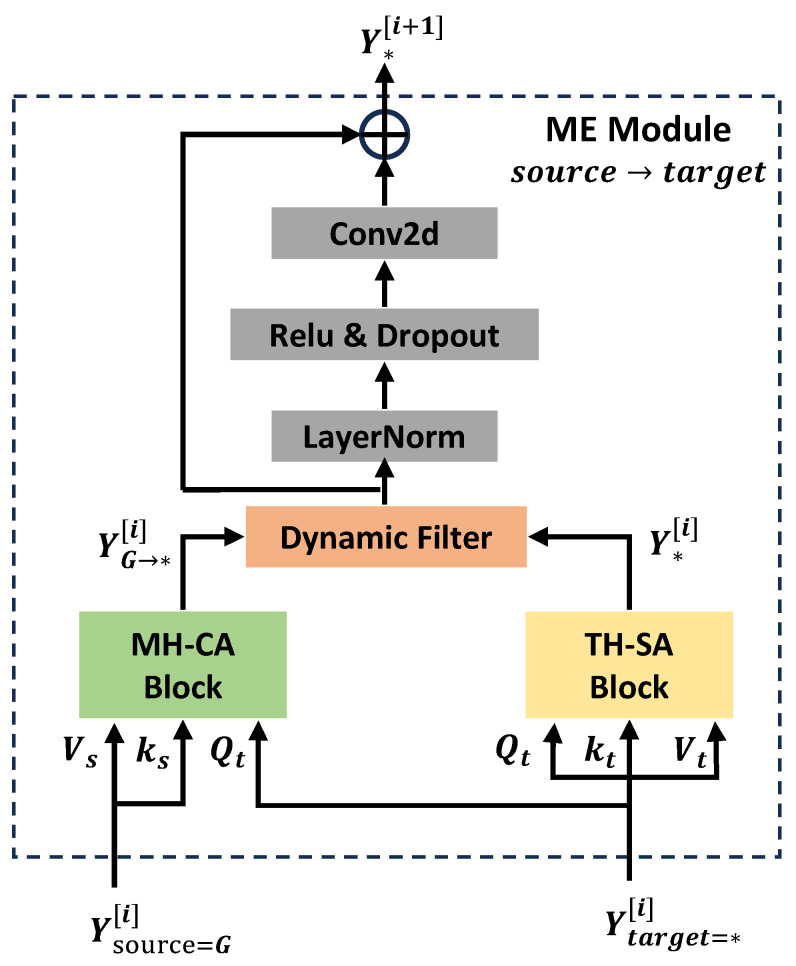
Illustration of the ME module. The ME module consists of a multi-head cross-modal attention(MH-CA) block and a two-person hypergraph self-attention (TH-SA) block. The skeleton features are gradually enhanced from the source modality to the target modality.

**Figure 3 biomimetics-09-00123-f003:**
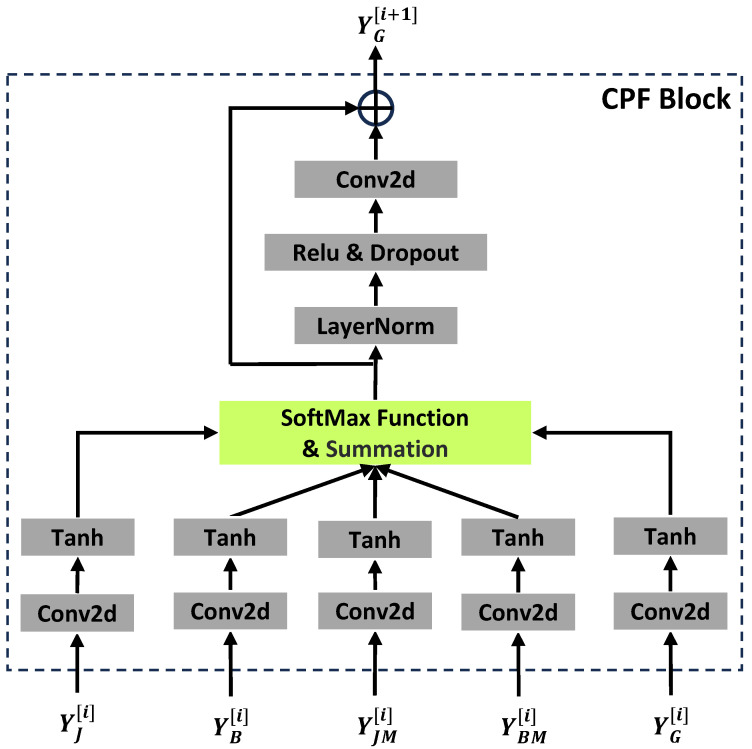
Illustration of the CPF block. This converts the features of four skeleton modalities and the previous global context into a higher-level global context for enhancing four single skeleton modalities in the next layer.

**Figure 4 biomimetics-09-00123-f004:**
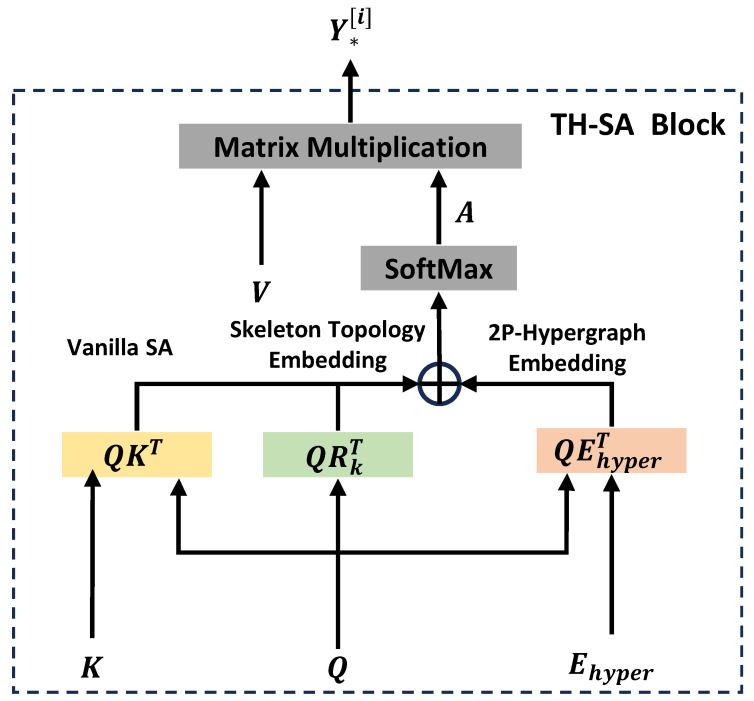
Illustration of the two-person hypergraph self-attention block (TH-SA). In addition to vanilla SA, this adds skeleton topology embedding and 2p-hypergraph embedding, allowing the transformer to better recognize the spatial structure and semantic information specific to each human joint.

**Figure 5 biomimetics-09-00123-f005:**
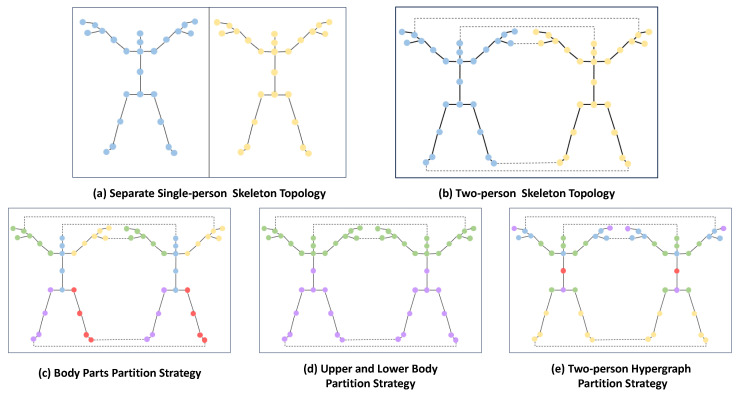
A brief visualization of different skeleton topologies and partition strategies. (**a**,**b**) are two kinds of skeleton topologies that contain different pairwise relations of skeleton joints. (**c**–**e**) are partition strategies that depict the higher-order relations of joints in a two-person skeleton topology. Different colors represent different higher-order relations. Joints with the same higher-order relation are set into a group and the entire two-person skeleton topology is divided into five groups. Multiple joints with the same high-order relation are labeled with the same color.

**Figure 6 biomimetics-09-00123-f006:**
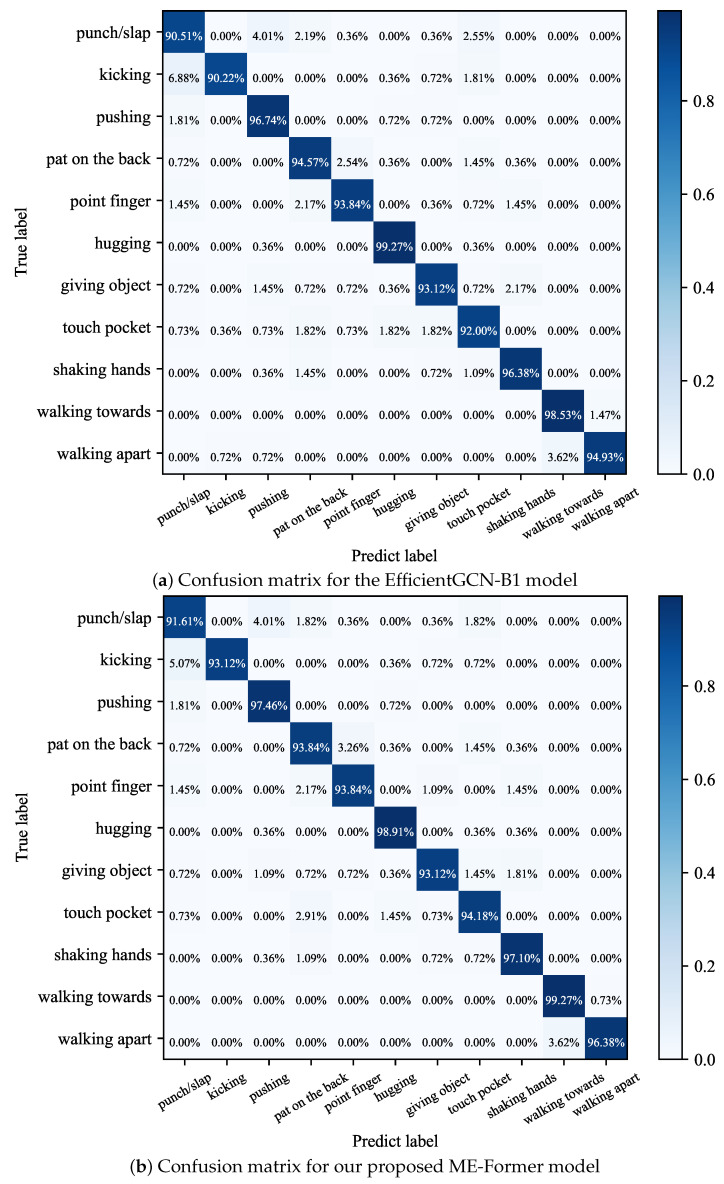
Confusion matrix diagram for two models testing on the NTU-60* (X-Sub) benchmark.

**Table 1 biomimetics-09-00123-t001:** Comparisons of interaction recognition performance on NTU-60* and NTU-120* datasets. The recognition accuracy of our proposed ME-Former are listed in the last row.

Type	Method	Conf./Jour.	NTU-60* (%)		NTU-120* (%)	
			**X-Sub**	**X-View**	**X-Sub**	**X-Set**
RNN	ST-LSTM [50]	TPAMI 2017	83.00	87.30	63.00	66.60
	GCA-LSTM [51]	CVPR 2017	85.90	89.00	70.60	73.70
	2s-GCA-LSTM [52]	TIP 2017	87.20	89.90	73.00	73.30
	FSNET [53]	TPAMI 2019	74.01	80.50	61.22	69.70
	GeomNet [54]	ICCV 2021	93.62	96.32	86.49	87.58
	LSTM-IRN [41]	TMM 2022	90.50	93.50	77.70	79.60
GCN	ST-GCN [31]	AAAI 2018	89.31	93.72	80.69	80.27
	AS-GCN [55]	CVPR 2019	89.30	93.00	82.90	83.70
	2s-AGCN [13]	CVPR 2019	93.36	96.67	87.83	89.21
	ST-GCN-PAM [13]	ICIP 2020	-	-	83.28	88.36
	Pa-ResGCN-B19 [56]	ACM MM 2020	94.34	97.55	89.64	89.94
	CTR-GCN [32]	ICCV 2021	91.60	94.30	83.20	84.40
	DR-GCN [40]	PR 2021	93.68	94.09	85.36	84.49
	K-GCN [39]	NC 2021	93.70	96.80	-	-
	2S-DRAGCN [40]	PR 2021	94.68	97.19	90.56	90.41
	AIGCN [43]	ICME 2022	93.89	97.22	87.80	87.96
	EfficientGCN-B1 [16]	TPAMI 2022	94.49	97.23	90.64	90.21
	2s-AIGCN [43]	ICME 2022	95.34	98.00	90.71	90.65
Transformer	DSTA-NET [57]	ACCV 2020	-	-	88.92	90.10
	IGFormer [42]	ECCV 2022	93.64	96.50	85.40	86.50
	STSA-Net [58]	NC 2023	-	-	90.28	91.13
	ISTA-Net [59]	IROS 2023	-	-	90.60	91.87
	**Our ME-Former**	-	**95.37**	**97.60**	**90.84**	**91.33**

**Table 2 biomimetics-09-00123-t002:** Ablation study on multi-modal enforcement transformer network.

Model Design		Params. (M)	NTU-60*X-Sub (%)
Feature Extraction and Fusion Stage	Feature Refinement Stage		
4 × SGC (Baseline)	SGC	1.2	94.49
4 × TH-SA	TH-SA	5.6	94.98
4 × ME	TH-SA	7.8	95.26
4 × (ME + CPF)	TH-SA	9.2	**95.37**

**Table 3 biomimetics-09-00123-t003:** Ablation study on components of TH-SA block.

Model Design	Params. (M)	NTU-60*X-Sub (%)
Vanilla SA	4.4	93.37
Vanilla SA + Single-person Skeleton Topology Embedding	4.8	94.34
Vanilla SA + Two-person Skeleton Topology Embedding	4.8	**94.55**
Vanilla SA + Two-person Hypergraph Embedding	5.1	94.67
Entire TH-SA	5.6	**94.98**

**Table 4 biomimetics-09-00123-t004:** Ablation study on partition strategies of TH-SA block.

Partition Strategy	NTU-60*X-Sub (%)
Upper and Lower Body	94.53
Body Parts	94.64
Two-person Hypergraph	**94.98**

## Data Availability

The datasets used in this paper are available online.

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
