# Peer review of "Multi-Modal Enhancement Transformer Network for Skeleton-Based Human Interaction Recognition"

_biomimetics, 2024, doi:10.3390/biomimetics9030123_

Round 1
Reviewer 1 Report
Comments and Suggestions for Authors
This manuscript as entitled Mutli-modal enhancement transformer network for skeleton based Human Interaction. I would recommend following addition in this manuscript.
1. Introduction section can be improved where authors explained their work, it could be clearer.
2. In Section 5.2: Can you please explain the rationale behind setting the batch size to 16 and resizing the frame number of a single sample to 64. How did you make a decision of using ten layers in your model, and why 192 per layer? Also, can you provide insights into the decision to use Stocahstic Gradient Descent for training along with Nesterov momentum set to 0.9 amd weight decay set to 0.0005?
3. you have effectively categorized the methods into LSTM-based nd GCN-based approaches. Can you please explain why these were choosen and how they represent different approaches to the problem.
Comments on the Quality of English Language
Fine
Reviewer 2 Report
Comments and Suggestions for Authors
The proposed manuscript is devoted to the description of a study of a new multi-modal enhancement transformer network for skeleton-based human interaction recognition. The proposed model can enhance the skeleton features of specific modal from other skeletal modalities and model spatial dependencies between joints using specific modality. The authors propose also a two-person skeleton topology and a two-person hypergraph representation.
Preliminaries to the research area are provided. In particular, recent human action recognition technologies and skeleton-based action recognition methods are reviewed. Novel deep-learning models in the field are described. Definitions of attention mechanism and hypergraph representation are briefly recapped.
The methodology of the proposed study is described in detail, in particular the data sets and implementation details are provided. Extensive experiments on benchmark NTU-RGB+D 60 and NTU-RGB+D 120 datasets are carried out. The characteristics of the proposed model are evaluated. Results of the experiments are provided. They are commented and compared with other approaches.
The proposed methodology shows quite good performance.
The presentation of the main results is clear and comprehensive. From a formal point of view, all the contents seems to be correct. The results are valuable and worthy of being published taking into account their possible applications in various fields such as bionic robots, medical care, video surveillance, intelligent transportation, human-robot interactions etc.
Minor revisions are suggested to improve the quality of the exposition:
p. 1, line 18: I suggest to write “NTU-RGB+D 60” instead of “NTU-RGB+D”.
p. 2, lines 49 and 76: Please check if there is no repetitions in these paragraphs, which could be avoided.
p. 3, line 106: I suggest a brief description of the content of the remaining Sections to be given at the end of the Introduction.
p. 8, line 260 and p. 10, lines 301 and 307: The formulae Eq. 9, Eq. 13 and Eq. 14 should end with period instead of comma.
p. 12, lines after 383: In Section 5.3 the proposed model is compared with other methods. It is mentioned that it is higher than others but it is not very clear with respect to what – the accuracy or smth. else. I suggest this to be clarified.
p. 16, line 505: I suggest the concluding section to be extended giving possible directions of development of the proposed approach.
Round 2
Reviewer 1 Report
Comments and Suggestions for Authors
Authors responded to all my comments as expected I have no more comments
Comments on the Quality of English Language
Fine